ᵃ | **Open Peer Review** | Antimicrobial Chemotherapy | Observation

# An antisense peptide-conjugated peptide nucleic acid (PPNA) for peptidoglycan recycling inhibition reduces AmpC hyperproduction and β–lactam resistance in *Pseudomonas aeruginosa*

Maria Escobar-Salom,[1,2,3] Isabel M. Barceló,[1,2,3] Jordi Sansó-Sastre,[1,2] Gabriel Torrens,[4] Elena Jordana-Lluch,[1,2,3] Bartolomé Moyà,[1,2,3] Antonio Oliver,[1,2,3] Carlos Juan[1,2,3]

**ABSTRACT** We performed a proof-of-concept study to validate a peptide-conjugated peptide nucleic acid (PPNA) directed to inhibit peptidoglycan recycling as strategy to reduce AmpC hyperproduction and β-lactam resistance in *Pseudomonas aeruginosa*. Our *nagZ*-targeting PPNA at 2 µM decreased mRNA levels of *nagZ* and *ampC* to about a quarter in the AmpC high-level hyperproducer mutant PAdacBΔD and a previously characterized clinical strain with similar features, causing low cytotoxicity on human A549 cells. Ceftazidime minimum inhibitory concentration decreased from 64 to 8 mg/L in both strains after combination with 2 µM PPNA (which showed significant synergy in checkerboard assays), suggesting that *nagZ*-targeting PPNAs can be explored as weapons to sensitize *P. aeruginosa* against β-lactams and return therapeutic value to these essential drugs.

**IMPORTANCE** In the current scenario of threatening antibiotic resistance rates in *Pseudomonas aeruginosa*, the quest for alternative therapeutic weapons must consider all options, including the use of antisense oligonucleotides (e.g., peptide-conjugated peptide nucleic acids [PPNAs]) to silence the production of key target proteins. In this regard, we designed a proof-of-concept study to validate a PPNA directed to inhibit peptidoglycan recycling as a strategy to impair AmpC β-lactamase hyperproduction and derived resistance in *P. aeruginosa*. Our results indicate that the designed PPNA (targeting the N-acetyl-glucosaminidase NagZ) at low concentrations significantly decreased AmpC production and ceftazidime resistance in clinically relevant high-level hyperproducer *P. aeruginosa* strains, suggesting interesting therapeutic potentials.

**KEYWORDS** *Pseudomonas aeruginosa*, AmpC β-lactamase, peptidoglycan recycling, NagZ, peptide-conjugated peptide nucleic acid (PPNA), ceftazidime

Peptidoglycan recycling key actors, such as AmpG permease or NagZ N-acetyl-gluco-saminidase, are known to be essential for AmpC hyperproduction and mediated β-lactam resistance in *Pseudomonas aeruginosa in vitro* and *in vivo* (1–4). Moreover, deletion of these elements attenuates *P. aeruginosa* virulence in a murine model (probably through increased susceptibilities to cell-wall-targeting immunity caused by the alterations that recycling blockade entails), altogether suggesting important therapeutic potentials (4, 5). Antisense peptide-conjugated peptide nucleic acids (PPNAs) are synthetic N-(2-aminoethyl)-glycine-based oligomers linked to a peptide enabling its permeabilization (cell-penetrating peptide, CPP), intended to block the translation of target proteins by hybridizing with their corresponding mRNAs, causing

Address correspondence to Carlos Juan, carlos.juan@ssib.es.

The authors declare no conflict of interest.

See the funding table on p. 5.

their degradation. They are emerging as therapeutic alternatives against opportunistic multidrug-resistant pathogens, such as *P. aeruginosa* (6–8). In view of these facts, we made a proof-of-concept study assessing a PPNA as potential weapon to disable AmpC hyperproduction and derived resistance, and thus return value to β-lactams. Given the above-mentioned virulence-related implications of peptidoglycan recycling blockade, we decided to target this process (specifically *nagZ*, in accordance with relevant research on inhibitors of the encoded enzyme [1, 9–13]) instead of designing PPNAs to directly silence *ampC*.

Our NagZ-PPNA, designed following manufacturer PNABIO instructions, is complementary of PAO1 *nagZ* sequence, positions −9 to +3 ("A" from ATG codon defined as +1): [N→C termini: (RXR)$_4$XB-O-CATGAAAAGTCC, with R: arginine; X: 6-aminohexanoic acid; B: β-alanine; -O-: ethylene-glycol]. (RXR)$_4$ sequence was chosen as CPP, and XB-O as linker for conjugation with the PNA in accordance with previous studies (6–8, 14). Following habitual procedures (14), a negative control PPNA (Scr-PPNA) was used to discard nonspecific effects of the CPP and/or the entry of foreign genetic-like material into bacteria *per se*. The Scr-PPNA consisted of a scrambled sequence based on that of NagZ-PPNA, with two nucleotides changed in position to avoid an efficient specific binding, but respecting the total numbers of A, T, G, and C [N→C: (RXR)$_4$XB-O-CTAAA AAGGTCC]. An additional negative control (Ctrl-PPNA) consisting of an 11-mer random sequence [N→C: (RXR)$_4$XB-O-CTGAGCACGAC] previously shown to display no significant impacts on *P. aeruginosa* viability (14) was also used to ensure the absence of nonspecific effects. All PPNAs had been characterized by the manufacturer in terms of purity (99.9%) by high-performance liquid chromatography (HPLC) and identity by matrix-assisted laser desorption/ionization mass spectrometry (MALDI MS) (Supplemental material).

To determine NagZ-PPNA minimum concentration valid to significantly reduce *nagZ* translation and consequently AmpC production without affecting viability in a well-known high-level hyperproducer mutant of clinical relevance (PAdacBΔD [3, 15]), we designed a protocol based on previous knowledge (16–19). Cells (5E$^4$) of the different strains were added to each well of a non-binding U bottom-shaped 96-well plate (Greiner Bio-One), in a final volume of 200 µL of non-adjusted Müller-Hinton broth containing decreasing concentrations of NagZ-PPNA, Ctrl-PPNA, or Scr-PPNA (16 to 0.125 µM). Plates were covered with gas-permeable sealing foils (Beckman) and incubated at 37°C-180 rpm agitation for 14 h, after which bacteria were harvested and their RNA purified to quantify *nagZ* and *ampC* expression by real-time RT-PCR, following published protocols and using regular negative controls to discard any DNA contamination or nonspecific amplification, and the well-known housekeeping gene *rpsL* to normalize results (20). RT-PCR primers for *ampC* and *rpsL* had been previously described (20), whereas those for *nagZ* were designed in this study (nagZ_F: CATGTCATCTATCCGCAGGTC; nagZ_R: GTCACC TTCAGGCGTTGCAG).

As shown in Fig. 1A and B, concentrations of NagZ-PPNA 8 to 2 µM had similar effects reducing the relative values of *ampC* and *nagZ* mRNAs in PAdacBΔD to ≈a quarter of the untreated strain, whereas a concentration-dependent decreasing effect in silencing was seen for 1 to 0.125 µM. Despite this NagZ-PPNA-mediated partial AmpC production blockade, this silencing did not reach that of the control used, the triple mutant PAdacBΔDnZ (defective in NagZ [2]), showing a reduction of ≈30-fold compared with PAdacBΔD *ampC* expression (Fig. 1A). The same experiments were performed with PAO1 wild type, and whereas comparable trends were seen for *nagZ* expression after exposition to NagZ-PPNA (Fig. 1C), no statistically significant changes were documented for *ampC* mRNA compared with no treatment (data not shown), probably because of the low expression in basal PAO1. Control incubations with Scr-PPNA or Ctrl-PPNA had no significant effects for the expression of *nagZ* or *ampC* in PAdacBΔD (Fig. 1A and B), strongly suggesting that the effects seen for NagZ-PPNA were target-specific. These assays were also used before RNA extraction to determine the minimum inhibitory concentrations (MICs) of NagZ-PPNA, Ctrl-PPNA, and Scr-PPNA (16 µM in all cases,

**TABLE 1** Minimum inhibitory concentrations (MICs) of PPNAs, ceftazidime (as indicator of β-lactam resistance profile), and combined treatments in the indicated *P. aeruginosa* strains

| Strain | MIC[a] | | | | | | |
|---|---|---|---|---|---|---|---|
| | NagZ-PPNA | Scr-PPNA | Ctrl-PPNA | CAZ | CAZ + NagZ-PPNA[b] | CAZ + Scr-PPNA | CAZ + Ctrl-PPNA |
| PAO1 | 16 | 16 | 16 | 1 | 1 | 1 | 1 |
| PAdacBΔD | 16 | 16 | 16 | 64 | 8 | 64 | 64 |
| PAdacBΔDnZ | ND | ND | ND | 2 | 2 | ND | ND |
| OFC2I4 | 16 | 16 | 16 | 64 | 8 | 64 | 64 |
| OFC2I4ΔnZ | ND | ND | ND | 2 | 2 | ND | ND |

[a]Results represent the median value obtained of three independent replicates. MICs of PPNAs are expressed in µM, whereas those of CAZ are in mg/L. Abbreviations: CAZ: ceftazidime; ND: not determined.
[b]In the combined treatments, PPNAs were added at a fixed final concentration of 2 µM.

Table 1), effects probably mediated by the permeabilization of bacterial membranes as previously described (14, 16, 19).

Therefore, 2 µM was chosen as the concentration of NagZ-PPNA used in combined treatments to ascertain whether the *ampC* expression reduction entailed a sensitization to β-lactams. Microdilution testing following the protocol indicated above was performed to determine MICs of ceftazidime alone or combined with PPNAs. Whereas Scr-PPNA and Ctrl-PPNA had no effect after combination with ceftazidime, 2 µM NagZ-PPNA decreased MICs from 64 to 8 mg/L in PAdacBΔD (Table 1), obviously not reaching the level of decrease of the control PAdacBΔDnZ (ceftazidime MIC = 2 mg/L). Interestingly, the level of sensitization caused by NagZ-PPNA was even higher than that achieved by a NagZ inhibitor (pUGNAC) previously tested against PAdacBΔD, for which a decrease of only half a dilution in ceftazidime MIC was seen (2). In accordance with previous results with the PAΔnZ mutant (showing similar susceptibility to wild type [2]), PAO1 ceftazidime MIC was not altered after NagZ-PPNA treatment and accordingly, this did not affect PAdacBΔDnZ either (Table 1). To complete our analysis, a previously characterized clinical strain showing a very similar profile to PAdacBΔD (AmpC hyperproduction mediated by *ampD* and *dacB* mutations), namely OFC2I4, and its derived nagZ-defective mutant (3) were used to reproduce the mentioned assays. As can be seen in Table 1 and Fig. 1D and E, results for this clinical strain were in fair accordance with those of PAdacBΔD both in terms of *nagZ* and *ampC* silencing and derived ceftazidime MICs, which increases the robustness of our approach. Incubation of OFC2I4 strain with Scr-PPNA or Ctrl-PPNA had no significant effects for *ampC* or *nagZ* expression (Fig. 1D and E), again strongly suggesting that our results with NagZ-PPNA are fairly target-specific.

To provide data further detailing the ceftazidime-sensitizing effects of NagZ-PPNA against AmpC hyperproducer strains, we performed a checkerboard assay following standard protocols (7) with PAdacBΔD as model. Fractional inhibitory concentration (FIC) indexes were calculated as previously described (7), and as can be seen in Fig. 1F, the synergistic combinations (FIC index < 0.5) were those consisting of ceftazidime 8 mg/L plus NagZ-PPNA 2 µM and ceftazidime 4 mg/L plus NagZ-PPNA 4 µM.

To test whether NagZ-PPNA could have toxic effects on eukaryotic cells as previously described for different CPPs (likely mediated by membrane permeabilization [16, 19, 21]), which would handicap its therapeutic potential, we incubated confluent cultures of human lung line A549 in 96-well plates for 14 h with RPMI-1640 medium (Biowest) alone, or complemented with potassium dichromate (100 µM) (positive control for total cell death) (19), or growing concentrations of NagZ-PPNA (from 2 µM, the lowest PPNA concentration displaying synergistic effect, to 32 µM). The Cytotoxicity Detection Kit PLUS (Roche) was used to quantify cell death (20), and whereas basal level in RPMI-1640 reached a ≈2.5% (relative to the 100% caused by potassium dichromate), after NagZ-PPNA treatment at 2 and 4 µM, values of this parameter were ca. 6% and 10% respectively (Fig. 1G), similar or even better data than those of other PPNAs against different pathogens (16, 19, 21).

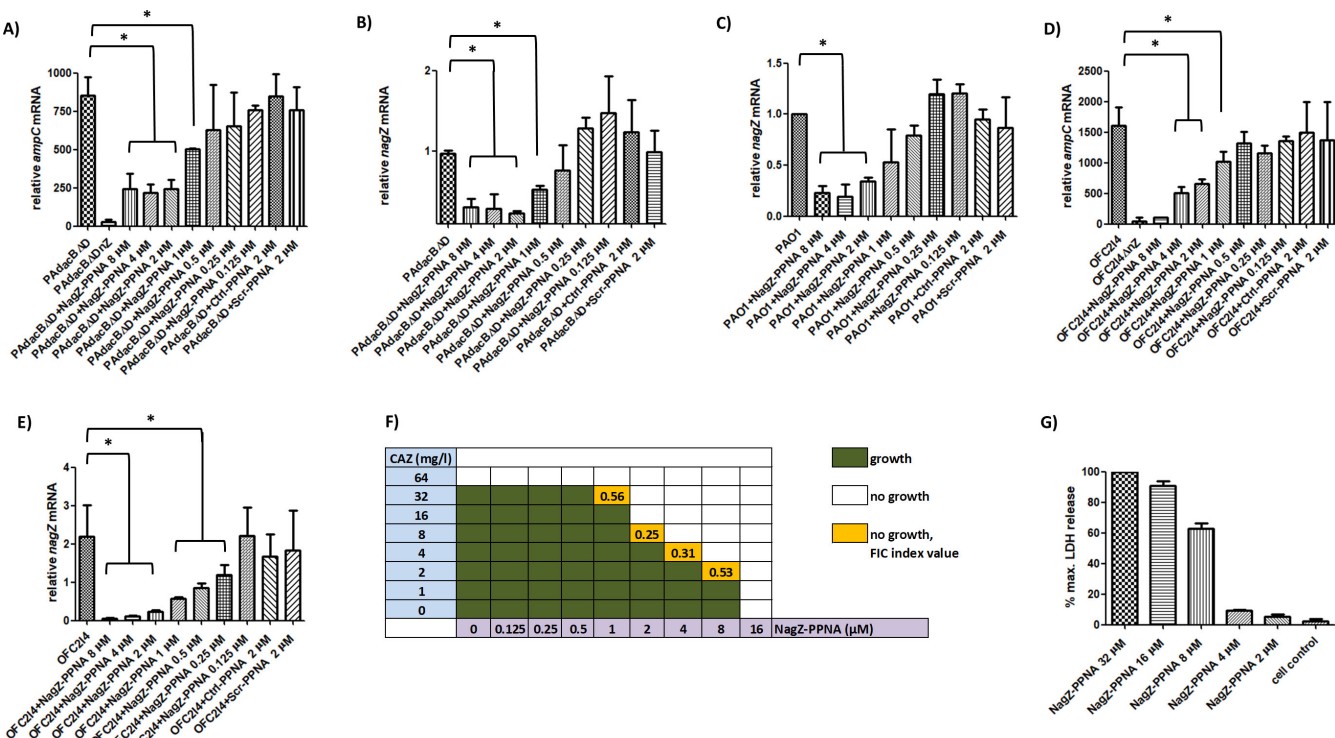

**FIG 1** Relative increase/decrease in the expression level of *ampC* and *nagZ* genes. Relative (fold) changes in mRNA of *ampC* (A, D) and *nagZ* (B, C, E) of the indicated strains, considering PAO1 expression as 1. (F) Checkerboard assay performed using the PAdacBΔD mutant and the indicated concentrations of each compound. Median values from three independent replicates are shown. (G) Cytotoxicity (LDH release) values after assays with A549 cells and the indicated concentration of NagZ-PPNA. Results are expressed as the percentages with regard to the maximum LDH that can be released, i.e., from a well of completely lysed confluent A549 cells. Columns represent the mean values from three independent biological replicates (each one consisting of three technical replicates), whereas the error bars correspond to the standard deviations. All data are displayed in linear scale. * indicates a *P*-value < 0.05 by ANOVA (Tukey's post hoc test) for multiple comparisons between the values indicated. Strains are grouped with horizontal lines when there was no statistical difference between them, whereas the symbols for obvious statistical significance have been omitted to declutter the figure.

Although we cannot discard a certain greater level of ceftazidime effectiveness caused by its CPP-mediated increased permeabilization (which would explain why with a high hyperproduction in treated PAdacBΔD [still ≈200-fold vs PAO1] its MIC was quite low [8 mg/L]), the effects of NagZ-PPNA seem quite target-specific as demonstrated by the NagZ-PPNA concentration-dependent overall increase in *nagZ* and *ampC* silencing (Fig. 1). Also, the fact that PPNAs did not increase PAO1 or PAdacBΔDnZ susceptibility to ceftazidime (Table 1) suggests that the unspecific effects could be minor. Microdilution performed in conditions slightly different from the standard ones (agitation, sealing foil, plates material, shorter incubation period) may also partially account for the mentioned ceftazidime susceptibility outputs.

Our study confirms, in accordance with previous evidence with mutants or NagZ inhibitors (1, 2, 9–13), that this is a valid target to reduce AmpC-dependent resistance in *P. aeruginosa*. Although ceftazidime sensitization through NagZ-PPNA was modest in the high-level hyper-producer mutant used, its effectiveness should be better in moderate AmpC hyperproducers such as single *ampD* or *dacB* mutants, as demonstrated for different NagZ inhibitors (1, 2, 11, 13). Another positive point is that we obtained significant effects with PPNA concentrations similar to or even lower than those showing effectiveness in other works with *P. aeruginosa* and other pathogens (6, 7, 16–19, 22). Higher concentrations of our PPNA may not be more effective because of some kind of saturation effect, as previously seen for other oligomers (23). Although our work is not the first that uses a PPNA to silence resistance genes (22–25), it is the unique targeting of a *P. aeruginosa* gene that besides antibiotic resistance, may impact virulence

(4). A limitation of our study is that for the moment, we have not performed animal models of infection to validate our *in vitro* data, and thus future work in this direction is needed to determine whether a significant attenuation of resistance and virulence can be achieved *in vivo* through our strategy. Besides, future research to assess different types of antisense oligonucleotides (e.g., phosphorodiamidate morpholino oligomers [14]), hybridization sites (in *nagZ*, but also in other peptidoglycan recycling key elements such as AmpG [3–5]) and different types of CPPs (14) will likely allow PPNA designs with improved therapeutic potentials to rehabilitate β-lactams effectiveness.

## ACKNOWLEDGMENTS

This work was supported by the Balearic Islands Government grant FPI/2206/2019 and grants IJC2019-038836-I and CNS2023-144168 (Ministerio de Ciencia, Innovación y Universidades, Spain), PI21/00753, PI24/00010, FI19/00004, and Centro de Investigación Biomédica en Red-Enfermedades Infecciosas (CB21/13/00099) from the Instituto de Salud Carlos III (Spain) co-financed by the European Regional Development Fund "A way to achieve Europe."

## AUTHOR AFFILIATIONS

[1]ARPBIG group, Health Research Institute of the Balearic Islands (IdISBa), Palma, Spain
[2]Microbiology Department, University Hospital Son Espases (HUSE), Palma, Spain
[3]Centro de Investigación Biomédica en Red, Área Enfermedades Infecciosas (CIBERINFEC), Instituto de Salud Carlos III (ISCIII), Madrid, Spain
[4]Department of Molecular Biology and Laboratory for Molecular Infection Medicine Sweden (MIMS), Umeå Centre for Microbial Research (UCMR), Umeå University, Umeå, Sweden

## AUTHOR ORCIDs

Elena Jordana-Lluch  http://orcid.org/0000-0002-1115-2048
Antonio Oliver  http://orcid.org/0000-0001-9327-1894
Carlos Juan  http://orcid.org/0000-0002-1402-3516

## FUNDING

| Funder | Grant(s) | Author(s) |
| --- | --- | --- |
| Govern de les Illes Balears | FPI/2206/2019 | Maria Escobar-Salom |
| Ministerio de Ciencia, Innovación y Universidades | IJC2019-038836-I | Elena Jordana-Lluch |
| Ministerio de Ciencia, Innovación y Universidades | CNS2023-144168 | Carlos Juan |
| Instituto de Salud Carlos III | PI21/00753 | Carlos Juan |
| Instituto de Salud Carlos III | PI24/00010 | Antonio Oliver |
| Instituto de Salud Carlos III | FI19/00004 | Isabel M. Barceló |
| Instituto de Salud Carlos III | CB21/13/00099 | Antonio Oliver |

## ADDITIONAL FILES

The following material is available online.

### Supplemental Material

**Supplemental material (Spectrum02622-24-S0001.docx).** Certificate of analysis of the PPNAs used in this study, provided by the manufacturer.

## Open Peer Review

**PEER REVIEW HISTORY (review-history.pdf).** An accounting of the reviewer comments and feedback.

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
