## [Reviewer comments · Microbiology Spectrum]

Microbiology Spectrum

An antisense Peptide-conjugated Peptide Nucleic Acid (PPNA) for peptidoglycan recycling inhibition reduces AmpC hyperproduction and β -lactam resistance in *Pseudomonas aeruginosa*

Maria Escobar-Salom, Isabel Maria Barceló, Jordi Sansó-Sastre, Gabriel Torrens, ELENA JORDANA LLUCH, Bartolomé Moyà, Antonio Oliver, and Carlos Juan

Corresponding Author(s): Carlos Juan, Institut d'Investigació Sanitària de les Illes Balears

Review Timeline:

Submission Date:	October 17, 2024
Editorial Decision:	December 16, 2024
Revision Received:	March 11, 2025
Editorial Decision:	April 4, 2025
Revision Received:	May 20, 2025
Editorial Decision:	June 4, 2025
Revision Received:	June 30, 2025
Accepted:	July 1, 2025

Editor: Silvia Cardona

Reviewer(s): The reviewers have opted to remain anonymous.

Transaction Report:

DOI: <https://doi.org/10.1128/spectrum.02622-24>

Re: Spectrum02622-24 (An antisense Peptide-conjugated Peptide Nucleic Acid (PPNA) for peptidoglycan recycling inhibition reduces AmpC hyperproduction and β -lactam resistance in *Pseudomonas aeruginosa*)

Dear Dr. Carlos Juan:

Thank you for submitting your manuscript to Microbiology Spectrum. Two experts in the field have reviewed your article. Both reviewers found concerns that should be addressed before the study can be considered for publication again. Please note that both reviewers noted important shortcomings regarding cytotoxicity studies.

Their recommendations are provided below.

Revision Guidelines

Sincerely,
Silvia Cardona
Editor
Microbiology Spectrum

Reviewer #1 (Comments for the Author):

In the study, the authors attempt to modulate ampC production by blocking peptidoglycan recycling through inhibition of nagZ with a PPNA. They demonstrate that a nagZ PPNA could improve the MIC of ceftazidime from 64 ug/ml to 8 ug/ml. This study is relatively limited in the data that is presented and a number of questions remain to be answered to help inform how viable this approach could be in using antisense to interrupt peptidoglycan recycling.

1. As the authors show, although there is a dose-response, this response reaches a plateau at 8 um. It is unclear why this is the case and further experiments are needed. Did the authors test higher concentrations of PPNAs to see if they could further drive down expression?
2. The toxicity studies that were performed are inadequate. Toxicity at a single concentration is not helpful and a concentration range should be used. Only then can one get an initial sense of therapeutic index.
3. The use of antisense as an adjuvant approach while intellectually interesting is fraught with a number of obstacles to becoming a reality including the problem of dosing two drugs to reach a clinical benefit. These limitations should be discussed. The absence of proof of concept animal data showing that this approach could be viable is a weakness.
4. It would have been stronger if the authors had tested this PPNA in a non-modified clinical isolate that has high ampC expression. Has this been performed?

Reviewer #2 (Comments for the Author):

The following experiments must be included:

1. p. 5 line 127: Cytotoxicity at 2 uM is not relevant. Cytotoxicity must be made as dose response up to at least 50 uM, and the data included in the manuscript
 2. Table 1. One concentration of the PNA is not sufficient for proper characterization of the synergy effect. A checkerboard experiment must be done.
- Fig. 1: This key experiment seems only supported by technical replicates. It must biological replicates.

Other points:

p3 lines 76 & 80: PNAs are peptides and do not have 3' and 5' ends

Reviewer #1 (Comments for the Author):

In the study, the authors attempt to modulate ampC production by blocking peptidoglycan recycling through inhibition of nagZ with a PPNA. They demonstrate that a nagZ PPNA could improve the MIC of ceftazidime from 64 ug/ml to 8 ug/ml. This study is relatively limited in the data that is presented and a number of questions remain to be answered to help inform how viable this approach could be in using antisense to interrupt peptidoglycan recycling.

1. As the authors show, although there is a dose-response, this response reaches a plateau at 8 μ M. It is unclear why this is the case and further experiments are needed. Did the authors test higher concentrations of PPNA to see if they could further drive down expression?

Regarding the mentioned plateau at 8 μ M, we believe that probably a saturation-type situation is reached here, because at this dosage no significantly improved entrance (and/or binding to target mRNA) of PNA may be achieved compared to 4 or 2 μ M. In fact, similar plateau-like results have been obtained in other studies, in which for instance: i) a PNA at 8, 16, and 32 μ M had virtually the same capacity to reduce *K. pneumoniae* biofilm mass to \approx a half of the controls (PMID: 29506074); ii) another PNA at 3 vs 12 μ M had very similar capacity to synergize with clindamycin at low concentrations against *E. coli* (PMID: 27631336), or iii) another PNA that showed virtually the same effect between 1 and 5 or 5 and 10 μ M to reduce pre-existing *P. aeruginosa* biofilms in different conditions (PMID: 28137807).

On the other hand, since our NagZ-PNA already showed *P. aeruginosa* growth inhibitory activity at 16 μ M (and the same for our negative control PNA, Table 1), we believe that could be not very informative to test higher concentrations, because the corresponding activity would not be target-specific but likely due to the effect of permeabilization itself. Another reason that supports that testing higher concentrations of PNA would not be very relevant is the toxicity for eukaryotic cells, which was very high already at 8 μ M (see below).

2. The toxicity studies that were performed are inadequate. Toxicity at a single concentration is not helpful and a concentration range should be used. Only then can one get an initial sense of therapeutic index.

Following the Reviewers' wise suggestion, we have performed new A549 toxicity studies with a wider range of PNA concentrations (32 to 2 μ M), which confirmed that at low doses (e.g. 2 and 4 μ M), our NagZ-PNA displayed relevant therapeutic activities without causing dramatic noxious impacts for the eukaryotic cells. In fact,

cytotoxicity values derived from the new set of experiments performed ad hoc for this revised version (regarding both control A549 wells and 2µM NagZ-PNA wells), were lower than those in the initial manuscript. Data have been added in the revised version (Fig 1).

3. The use of antisense as an adjuvant approach while intellectually interesting is fraught with a number of obstacles to becoming a reality including the problem of dosing two drugs to reach a clinical benefit. These limitations should be discussed. The absence of proof of concept animal data showing that this approach could be viable is a weakness.

We have added some lines discussing the mentioned limitation of our study, following the suggestion of the Reviewer.

4. It would have been stronger if the authors had tested this PPNA in a non-modified clinical isolate that has high ampC expression. Has this been performed? Following the wise recommendation of the Reviewer, we introduced in the study our previously characterized clinical isolate OFC214 (PMIDs: 16251318, 21357303) showing high level AmpC hyperproduction linked to *ampD* and *dacB* mutations. As can be seen in the revised version, results regarding ceftazidime MICs reduction with this strain are in fair accordance with the data using our *ampD-dacB*-defective PAO1 mutant shown in the first version of the manuscript.

Reviewer #2 (Comments for the Author):

The following experiments must be included:
1. p. 5 line 127: Cytotoxicity at 2 uM is not relevant. Cytotoxicity must be made as dose response up to at least 50 uM, and the data included in the manuscript

As wisely suggested by the reviewer, we tested different growing PNA concentrations against A549 cells; although high cytotoxic effects were seen for NagZ-PNA already at 8 µM, it is interesting that at concentrations showing clear therapeutic power sensitizing *P. aeruginosa* against ceftazidime (e.g. 2 and 4 µM), the cytotoxic effects were quite low (up to 10% of cellular death). New data in this regard have been included in the revised version as part of Figure 1. Interestingly, cytotoxicity values derived from the new set of experiments performed ad hoc for this revised version (regarding controls and 2µM NagZ-PNA A549 wells), were lower than those in the first manuscript.

2. Table 1. One concentration of the PNA is not sufficient for proper characterization of the synergy effect. A checkerboard experiment must be done.

Checkerboard assays have been performed and results included in the revised version. Interestingly, clear synergistic effects have been found at different CAZ-PNA combinations.

Fig. 1: This key experiment seems only supported by technical replicates. It must biological replicates.

We apologise for the confusing methodological explanation in the initial version of the manuscript. In fact, results correspond to three independent biological replicates, each of them consisting of 3 technical replicates. We have corrected this explanation in the revised version.

Other points:

p3 lines 76 & 80: PNAs are peptides and do not have 3' and 5' ends

It has been corrected according to the observation of Reviewer, and renamed with "N and C termini" (PMID: 33921011).

Re: Spectrum02622-24R1 (An antisense Peptide-conjugated Peptide Nucleic Acid (PPNA) for peptidoglycan recycling inhibition reduces AmpC hyperproduction and β -lactam resistance in *Pseudomonas aeruginosa*)

Dear Dr. Carlos Juan:

Thank you for submitting your revised manuscript to Microbiology Spectrum. Your article has been reviewed by two experts in the field. One reviewer requires clarification on some of your controls. I agree with the reviewer and ask you to consider the requests before making any changes. Although this manuscript is in the form of an Observation, you still need to provide the information to evaluate that the experimental approach is sound.

Selected reviewer's recommendations are provided below:

Revision Guidelines

Sincerely,
Silvia Cardona
Editor
Microbiology Spectrum

1) Was chemical characterization of the anti-nagZPPNA, in terms of purity (HPLC) and identity (mass spec analysis) performed? please clarify

2) Table 1 and Figure 1 should include a sequence control PNA to demonstrate a target hybridization-dependent activity. It is unclear which control was used as "Ctrl-PPNA" is not defined in the text. Please explain the relevance of this control and whether it is related to the anti-nagZPPNA. A reviewer has suggested that better control would be to interchange two bases in the nagZPPNA: CATAAAAGGTCC, as this would create two mismatches with the intended target and abolish binding. Explain how the difference between your control and the one suggested by the reviewer could affect the interpretation of your results. Acknowledge any limitations of your experimental design.

3) What is the internal control of the RT-PCR experiments ? Clarify this in the text/

4) Indicate whether the peptide used could exhibit toxicity. A reviewer states that cationic delivery peptides, and especially arginine-rich ones, exhibit toxicity towards both (gram-negative) bacteria as well as eukaryotic cells.

RESPONSE TO REVIEWERS

First of all, we thank Reviewers and Editor for their wise observations. Moreover, we respectfully request the possibility of adding two new co-authors now; we forgot to do so in the first revision round (we apologize for that), although they had a crucial role to make the experiments relative to both first and 2nd revisions. Obviously, if this is not possible, we will maintain the same authors we formerly had.

1) Was chemical characterization of the anti-nagZPPNA, in terms of purity (HPLC) and identity (mass spec analysis) performed? please clarify

The manufacturer (PNABio, USA) documents sent to us together with the product indicate that PPNA had been characterized in terms of purity (shown to be 99.9%) by HPLC (Agilent 1100 Series device) and identity by MALDI MS (AXIMA-Assurance device, Shimadzu Biotech), providing the following MS data: NagZ-PPNA: 5297.4 m/z; MS Ctrl-PPNA: 5021.8 m/z.

A sentence in the text has been added in this regard.

2) Table 1 and Figure 1 should include a sequence control PNA to demonstrate a target hybridization-dependent activity. It is unclear which control was used as "Ctrl-PPNA" is not defined in the text. Please explain the relevance of this control and whether it is related to the anti-nagZPPNA. A reviewer has suggested that better control would be to interchange two bases in the nagZPPNA: CATAAAAGGTCC, as this would create two mismatches with the intended target and abolish binding. Explain how the difference between your control and the one suggested by the reviewer could affect the interpretation of your results. Acknowledge any limitations of your experimental design.

We included in the Figure 1 the requested control. In other words, the AmpC hyperproducer strains used in this study (PAdacBAD and OFC214) were incubated in presence of 2 μ M of Ctrl-PPNA (a concentration showing high ampC and nagZ silencing activity for the NagZ-PPNA), and afterwards RNA was extracted to determine the level of ampC and nagZ expression (RT-PCR). As can be seen in

the new Figure 1, results prove that the effects seen for NagZ-PPNA were target-specific, and that Ctrl-PPNA has no effects.

We believe that the Ctrl-PPNA used was valid, since its sequence was obtained from a recently published article (PMID: 37610213), in which it was shown to have no important effects on viability of *P. aeruginosa* (MIC > 16 mg/l) and described to be a 11-mer random sequence not complementary to essential targets. Therefore we do believe that is very unlikely that this previously described oligomer could have effects driving to artifacts in our study. However, we cannot discard some marginal effect of Ctrl-PPNA (or of NagZ-PPNA) affecting *P. aeruginosa* not by specifically hybridizing, but through permeabilization exerted by the cell-penetrating conjugated peptide.

We have emphasized all these circumstances in the text.

3) What is the internal control of the RT-PCR experiments ? Clarify this in the text/ Besides the control posed by RNAs extracted after incubation with the Ctrl-PPNA (see previous point), our internal control is the house keeping gene *rpsL*, which has been used to normalize RT_PCR results in several studies with *P. aeruginosa*. We have very large experience using this gene in our laboratory, and therefore we know which CT values are within normal results. Moreover, negative controls in which no RNA is added to the RT_PCR tubes (to discard any inespecific result of Amplification/fluorescence, or contamination of reagents with positive samples) and others in which no retrotranscriptase is added to the mix (to discard that samples are contaminated with DNA) were routinely performed to validate results.

Moreover, we do believe that incubation with Ctrl-PPNA driving to no measurable changes in *ampC* or *nagZ* expression poses also an appropriate control to demonstrate that our effects with NagZ-PPNA were target specific.

We have emphasized all these circumstances in the text.

4) Indicate whether the peptide used could exhibit toxicity. A reviewer states that cationic delivery peptides, and especially arginine-rich ones, exhibit toxicity towards both (gram-negative) bacteria as well as eukaryotic cells. We have added a sentence in the text to emphasize that cell-penetrating peptides have been described to have certain levels of eukaryotic cells toxicity (likely mediated by permeabilisation), which is a well known circumstance in studies with PPNA. In fact, it has been shown that PNAS conjugates with CPP consisting of

(RXR)4 (as those we used in our assays) are sometimes more toxic for eukaryotic cells than other CPPs such as (R-Ahx)6 or (KFF)3K (<https://doi.org/10.1089/nat.2012.0370>; <https://doi.org/10.3390/biomedicines9040429>). In any case, our toxicity results are relevant because at concentrations displayed to have therapeutic power against *P. aeruginosa* (2 and 4 μ M), were barely toxic for A549 cells.

We have emphasized all these circumstances in the text.

Re: Spectrum02622-24R2 (An antisense Peptide-conjugated Peptide Nucleic Acid (PPNA) for peptidoglycan recycling inhibition reduces AmpC hyperproduction and β -lactam resistance in *Pseudomonas aeruginosa*)

Dear Dr. Carlos Juan:

Your article was reviewed, and one reviewer still has concerns with your control PNA. I have consulted with other editors in Spectrum, and they agree you should address this reviewer's concern. If you feel you can do it, please follow the instructions below.

Revision Guidelines

Sincerely,
Silvia Cardona
Editor
Microbiology Spectrum

Reviewer #2 (Comments for the Author):

Unfortunately, two issues remain:

Since the two PNAs used are NOT a standard product that can be traced for identity and purity, the data showing this must be included in the paper. Thus the authors must either themselves obtain purity data (HPLC traces) and mass spectrometry data or

obtain these from the supplier.

This data must be available in supplementary.

Also, the control PNA used is not adequate as it is not dedicated and fully relevant for the designed antisense PNA. Indeed, it is one nucleobase shorter (11 nucleobases instead of 12) and the base composition (A3G3C4T) is different than that of the active PNA: A5G2C3T2.

A proper control is chemically as close as possible to the parent compound, except that it should not efficiently bind the nucleic acid target.

The control used is not a control it is just another (inactive) PNA, and using the author's argumentation, this would be a valid control for any PNA (having the same delivery peptide).

RESPONSE TO REVIEWERS

Reviewer #2 (Comments for the Author):

Unfortunately, two issues remain:

Since the two PNAs used are NOT a standard product that can be traced for identity and purity, the data showing this must be included in the paper. Thus the authors must either themselves obtain purity data (HPLC traces) and mass spectrometry data or obtain these from the supplier.

This data must be available in supplementary.

Also, the control PNA used is not adequate as it is not dedicated and fully relevant for the designed antisense PNA. Indeed, it is one nucleobase shorter (11 nucleobases instead of 12) and the base composition (A3G3C4T) is different than that of the active PNA: A5G2C3T2.

A proper control is chemically as close as possible to the parent compound, except that it should not efficiently bind the nucleic acid target.

The control used is not a control it is just another (inactive) PNA, and using the author's argumentation, this would be a valid control for any PNA (having the same delivery peptide).

We have included the HPLC and MS analysis of all the used PPNA provided by the supplier as supplementary material, following the request of the reviewer.

As suggested by the reviewer, we designed a new negative control PPNA (Scr-PPNA), based on the sequence of NagZ-PPNA, just changing two nucleotides of position but respecting the length and proportions of A,G,C,T of the original. We repeated all the experiments including this control, and as expected, no nonspecific effects were seen.

Re: Spectrum02622-24R3 (An antisense Peptide-conjugated Peptide Nucleic Acid (PPNA) for peptidoglycan recycling inhibition reduces AmpC hyperproduction and β -lactam resistance in *Pseudomonas aeruginosa*)

Dear Dr. Carlos Juan:

Thank you for your work addressing the reviewers' concerns. Your manuscript has been accepted, and I am forwarding it to the ASM production staff for publication. Your paper will first be checked to make sure all elements meet the technical requirements. ASM staff will contact you if anything needs to be revised before copyediting and production can begin. Otherwise, you will be notified when your proofs are ready to be viewed.

Sincerely,
Silvia Cardona
Editor
Microbiology Spectrum